# Comparison of Pulsed Radiofrequency, Oxygen-Ozone Therapy and Epidural Steroid Injections for the Treatment of Chronic Unilateral Radicular Syndrome

**DOI:** 10.3390/medicina57020136

**Published:** 2021-02-04

**Authors:** Pavel Ryska, Jiri Jandura, Petr Hoffmann, Petr Dvorak, Blanka Klimova, Martin Valis, Milan Vajda

**Affiliations:** 1Department of Diagnostic Radiology, University Hospital Hradec Kralove, Sokolska 581, 50005 Hradec Kralove, Czech Republic; pavel.ryska@fnhk.cz (P.R.); jiri.jandura@fnhk.cz (J.J.); petr.hoffmann@fnhk.cz (P.H.); petr.dvorak@fnhk.cz (P.D.); milan.vajda@fnhk.cz (M.V.); 2Department of Diagnostic Radiology, Faculty of Medicine in Hradec Kralove, Charles University, Simkova 870, 50003 Hradec Kralove, Czech Republic; 3Department of Neurology, University Hospital Hradec Kralove, Sokolska 581, 50005 Hradec Kralove, Czech Republic; martin.valis@fnhk.cz

**Keywords:** pulsed radiofrequency treatment, ozone, epidural injections, low back pain

## Abstract

*Background and objectives:* For the treatment of chronic unilateral radicular syndrome, there are various methods including three minimally invasive computed tomography (CT)-guided methods, namely, pulsed radiofrequency (PRF), transforaminal oxygen ozone therapy (TFOOT), and transforaminal epidural steroid injection (TFESI). Despite this, it is still unclear which of these methods is the best in terms of pain reduction and disability improvement. Therefore, the purpose of this study was to evaluate the short and long-term effectiveness of these methods by measuring pain relief using the visual analogue scale (VAS) and improvement in disability (per the Oswestry disability index (ODI)) in patients with chronic unilateral radicular syndrome at L5 or S1 that do not respond to conservative treatment. *Materials and Methods:* After screening 692 patients, we enrolled 178 subjects, each of whom underwent one of the above CT-guided procedures. The PRF settings were as follows: pulse width = 20 ms, f = 2 Hz, U = 45 V, Z ˂ 500 Ω, and interval = 2 × 120 s. For TFOOT, an injection of 4–5 mL of an O_2_-O_3_ mixture (24 μg/mL) was administered. For the TFESI, 1 mL of a corticosteroid (betamethasone dipropionate), 3 mL of an anaesthetic (bupivacaine hydrochloride), and a 0.5 mL mixture of a non-ionic contrast agent (Iomeron 300) were administered. Pain intensity was assessed with a questionnaire. *Results:* The data from 178 patients (PRF, n = 57; TFOOT, n = 69; TFESI, n = 52) who submitted correctly completed questionnaires in the third month of the follow-up period were used for statistical analysis. The median pre-treatment visual analogue scale (VAS) score in all groups was six points. Immediately after treatment, the largest decrease in the median VAS score was observed in the TFESI group, with a score of 3.5 points (a decrease of 41.7%). In the PRF and TFOOT groups, the median VAS score decreased to 4 and 5 points (decreases of 33% and 16.7%, respectively). The difference in the early (immediately after) post-treatment VAS score between the TFESI and TFOOT groups was statistically significant (*p* = 0.0152). At the third and sixth months after treatment, the median VAS score was five points in all groups, without a statistically significant difference (*p* > 0.05). Additionally, there were no significant differences in the Oswestry disability index (ODI) values among the groups at any of the follow-up visits. Finally, there were no significant effects of age or body mass index (BMI) on both treatment outcomes (maximum absolute value of Spearman’s rank correlation coefficient = 0.193). *Conclusions:* Although the three methods are equally efficient in reducing pain over the entire follow-up, we observed that TFESI (a corticosteroid with a local anaesthetic) proved to be the most effective method for early post-treatment pain relief.

## 1. Introduction

Minimally invasive treatment methods are important therapeutic options for patients suffering from chronic lumbar radicular syndrome, and these methods involve the use of computed tomography (CT) or fluoroscopy to ensure accurate needle placement and to prevent adverse reactions [1,2].

Three methods that are commonly used to treat lumbar radicular syndrome are epidural steroid injection (ESI), pulsed radiofrequency (PRF), and oxygen-ozone therapy (OOT). Epidural steroid injection (ESI) of corticosteroids is one of the most commonly used minimally invasive interventions for the management of chronic spinal pain [3]. Corticosteroids are known to affect processes associated with pain by inhibiting prostaglandin synthesis, impairing both the cell-mediated and humoral immune responses, stabilizing cellular membranes, and blocking nociceptive C-fibre conduction [3,4], and might also inhibit the formation of nerve root oedema [4]. Additionally, corticosteroids may be administered alone or in combination with other drugs, including local anaesthetics [5]. ESI is generally well tolerated, and most complications are related to technical problems. Nevertheless, rare cases of severe adverse reactions associated with ESI (such as spinal cord infarction) may occur [6,7,8].

Pulsed radiofrequency (PRF) was developed, in part, as a less invasive alternative to conventional continuous radiofrequency [9]. Although the exact mechanisms of action of PRF and its therapeutic effects remain unclear, it is assumed that the electric field generated by radiofrequency stimulation can promote microstructural changes in neural tissues, which in turn can block pain transmission [10]. Dorsal root ganglion (DRG) PRF also has an immunomodulatory effect, activates descending anti-nociceptive adrenergic and serotoninergic pathways, and significantly modulates microglial expression [11]. PRF is generally considered to be a very safe method [12].

Oxygen-ozone therapy (OOT) is based on the injection of an O_2_-O_3_ gas mixture, which can be applied intradiscally, to the epidural space or to the paravertebral muscle [13]. The proximal injection of the O_2_-O_3_ gas mixture to the root ganglion is thought to normalize the levels of cytokines and prostaglandins, increase superoxide dismutase levels, minimize reactive oxidant species levels, and improve local peri-ganglionic circulation, and this mixture has an eutrophic effect on the nerve root [14]. Complications of OOT can arise from the application technique used or adverse effects of the O_2_-O_3_ mixture [15,16]. However, the likelihood of complications from OOT is low [17].

In a previous retrospective study, Ding et al. [18] compared the efficacies of CT-guided transforaminal epidural steroid injection (TFESI), PRF and the combination of both methods in the treatment of lumbar disc herniation and observed a good early effect in the TFESI and TFESI + PRF groups. However, at three and six months after treatment, the reported visual analogue scale (VAS) and Oswestry disability index (ODI) scores were significantly higher in the TFESI group than in the PRF and combination groups. Furthermore, in a prospective comparative study, Lee et al. [19] examined the effectiveness of PRF and TFESI in relieving radicular pain due to disc herniation. The authors noted that in both study groups, the VAS scores for cervical and lumbar radicular pain significantly decreased at 12 weeks after treatment, and the ODI scores also decreased. No statistically significant differences in the VAS scores were observed between the PRF and TFESI groups at any time during the follow-up period. Additionally, in a double-blind randomized controlled study, Bonetti et al. [14] compared the efficacy of periradicular O_2_-O_3_ therapy to that of TFESI in patients with acute and chronic low back pain (LBP) and sciatic nerve pain. The authors reported complete pain remission in most patients, regardless of the type of treatment. At the six-month follow-up, the differences in favour of O_2_-O_3_ treatment were significant in patients with disc disease but not in those without disc disease.

PRF treatment is increasingly being used in the management of chronic pain syndromes [12,20]. The use of PRF in the DRG of the cervical radicular spine is promising. However, with regard to its lumbosacral counterpart, the use of PRF cannot be similarly advocated yet [12,21]. Shanthanna et al. [22] evaluated the efficacy of PRF treatment in the lumbar DRG in patients with chronic lumbar radicular pain in a triple-blind randomized controlled trial, with a placebo used in the control group. The authors found a relatively low level of success with PRF-DRG at the third month after treatment. The maximum difference in the VAS score was observed at 24 h, with a smaller difference at week 4. The mean VAS and ODI score differences did not significantly vary at four weeks or at three months.

These data indicate that there is still confusion about the choice of the most appropriate method for the treatment of chronic lumbar unilateral radicular syndrome. The choice of the method must be based on certain criteria including effectiveness in terms of pain reduction and improvement of disability. In this regard, to our knowledge, a comparison of these parameters among these three methods is not yet available. Thus, the aim of this study was to compare the efficacies of PRF, transforaminal oxygen ozone therapy (TFOOT), and transforaminal epidural steroid injection (TFESI) in relieving pain and improving disability in patients with chronic unilateral radicular syndrome. To this end, we used the same CT-guided minimally invasive approach to administer PRF, TFOOT, and TFESI and assessed the levels of pain relief (per the visual analogue scale (VAS)) and improvement in disability (per the Oswestry disability index (ODI)) of patients over a period of six months following treatment.

## 2. Materials and Methods

### 2.1. Patient Selection

This study was a prospective, randomized, single-blinded, monocentric trial and was approved by the Institutional Ethics Committee of the University Hospital Hradec Kralove (reference number: 201410 S06P, date of approval: 6 October 2014), and no registration was required according to the State Institute for Drug Control, as it does not evaluate a new active substance or any new therapies. The study was conducted in the Department of Radiology of the University Hospital Hradec Kralove in the Czech Republic. Patient recruitment started in December 2014 and ended in September 2017. The follow-up period for each patient was six months (questionnaire survey), and the final six-month follow-up dates were in March 2018.

A total of 692 patients were screened for eligibility, and 189 of them were deemed eligible. The enrolled patients were then randomized to one of the three treatment groups and given the indicated treatment. The data from 178 patients (PRF, n = 57, TFOOT, n = 69, TFESI, n = 52) were used for the statistical analysis at baseline, as 11 patients dropped out at the third month of the follow-up period. The data from 154 (PRF, n = 50, TFOOT, n = 57, TFESI, n = 47) patients were used for the analysis at the six month. A flow diagram showing the patient inclusion process, including reasons for exclusion, is shown in Figure 1.

### 2.2. Participants

A total of 178 patients with chronic unilateral radicular syndrome who were advised to undergo minimally invasive therapy by specialized clinicians (e.g., neurologists, neurosurgeons) were included in the study. The patients were informed of the purpose of the study, principles of treatment, possible complications, and post-treatment regimen. The possibility that the study may be interrupted due to the need for retreatment or another reason was emphasized. Written informed consent was required. The patients were randomly divided into the experimental groups and were blinded to the treatment group to which they were assigned. The study was conducted as part of regular shifts aimed at treating patients with back pain. Prior to each of these shifts, the order of execution of the individual methods was determined by lot (e.g., 1. PRF, 2. TFESI, 3. PRF, ...). Patients who met the criteria for inclusion in the study and agreed to be treated by the method according to the pre-drawn lot order were included. Thus, 0–5 patients were enrolled in the study during one shift.

Doctors were directly involved in the treatment of patients (P.R., J.J.). Personal data and data from the obtained questionnaires were recorded by persons not directly involved in the treatment (doctors P.D., P.H.).

The inclusion criteria were as follows: non-progressive, clearly dominant mono-segmental radicular pain in the unilateral dermatome L5 or S1, insufficient response to conservative therapy, the absence of indications for surgical treatment, a duration of pain of >3 months, a VAS score ≥ 3/10, morphologic correlate depicted by magnetic resonance imaging (MRI) or CT (e.g., disc protrusion, foraminal stenosis), international normalized ratio (INR) ≤ 1.2, and age ≥ 18 years. The exclusion criteria were as follows: leg paresis or paralysis; sphincter insufficiency; local infections or sepsis; oncologic diseases; haemorrhagic diathesis or anticoagulation (INR > 1.2); uncontrolled diabetes or other severe internal comorbidities; allergies to the medication or materials used; glucose-6-phosphate dehydrogenase deficiency; pregnancy; or uncooperativeness.

### 2.3. Equipment

For CT, the Somatom Definition AS + scanner (Siemens, Erlangen, Germany) was used. For PRF, the RFG˗1B radiofrequency generator (Cosman Medical, Burlington, MA, USA), a dispersive electrode, a thermocouple electrode, and a 22G SMK RF cannula (NeuroTherm, Amsterdam, The Netherlands) were used. For TFOOT, the OZO2 FUTURA ozone generator (Alnitec, Cremosano, Italy), 22G Spinocan needles (B Braun Medical, Melsungen, Germany), a 10 mL syringe, and a source of medical oxygen were used. For TFESI, 22G Spinocan needles, a 5 mL syringe, a corticosteroid solution (betamethasone dipropionate), a local anaesthetic solution (bupivacaine hydrochloride), and a non-ionic contrast agent solution (Iomeron 300) were used.

### 2.4. Interventions

All therapies were performed in an outpatient setting. Prior to the intervention, patients were instructed to expect different feelings in the area of the intervention site and in the treated lower limb during the needle insertion and treatment themselves, as well as to communicate these feelings to the medical staff on their regular inquiries. In all three groups (PRF, TFOOT, TFESI), the loading technique and size (22G) were identical. Feelings when inserting the same needle/radiofrequency (RF) cannula in patients are usually markedly individual. Similarly, in the case of the application of drug mixtures and stimulation and the actual effect of the RF treatment, the feelings of the treated patients (qualitatively and quantitatively) are not the same within the individual types of treatment. We therefore consider the possibility of unambiguous identification of the method by patients on the basis of emotion to be considerably limited and unreliable. Auditory perceptions were masked by music played in the CT scanner room. Visual identification was excluded by the patient’s position. During the procedure, the patients were placed in the prone position. The generators were set and the drug mixtures prepared outside of the view of the patients. The sounds of all the procedures were masked by music being played in the CT scanner room. CT navigation utilizing a low-dose protocol was used. The entry point and needle trajectory were identified by axial CT scan measurements. After aseptic preparation, the needle (RF cannula) was introduced with a few steps into the chosen neuroforamen, and each time the needle needed to be repositioned, the position was graphically verified.

For PRF, the dispersive electrode was placed on the thigh of the limb contralateral to the treated side. After proper RF cannula insertion, the stylet was replaced by the thermocouple electrode. The distance of the electrode tip position to the DRG was subsequently verified and corrected by sensory and motor nerve stimulation when possible. The patients were expected to confirm a change in feeling (e.g., tingling) to the sensory stimulation at a frequency of f = 50 Hz and voltage of U = 0.3–0.5 V. Similarly, a reaction to the motor stimulation was expected at a frequency of f = 2 Hz and voltage of U = 0.5–0.7 V. The PRF settings used were as follows: pulse width = 20 ms, f = 2 Hz, U = 45 V, Z ˂ 500 Ω, and interval = 2 × 120 s.

For TFOOT, the O_2_-O_3_ mixture was created by an ozone generator and transferred to a syringe. A total of 4–5 mL (24 μg/mL) of the O_2_-O_3_ mixture was injected gradually and slowly through the inserted needle. For TFESI, a mixture of a corticosteroid (1 mL), local anaesthetic (3 mL), and contrast agent (0.5 mL) was utilized. In the TFOOT and TFESI procedures, negative blood aspiration always preceded the injection. The distribution of the gas/drug mixtures used was imaged (Figure 2a,b). After the procedure, the patients sat for 30 min in the waiting room and then were free to leave if they were accompanied by another person. They were also advised to rest at home for the next two days.

### 2.5. Outcome Evaluation and Variables

Changes in pain intensity were quantified by means of the VAS (0–10 points), and patient disability was measured by means of the ODI (0–50 points). Before treatment and at 3 and 6 months after treatment, we used a printed questionnaire containing the VAS and the Czech translation of the Oswestry disability questionnaire, which addresses 10 aspects of daily life activities for the calculation of the ODI. The early (result within 30 min after treatment) post-treatment VAS score of each patient was obtained via interview after the procedure.

The patients were randomly assigned to the treatment groups, and the data from the questionnaire were recorded by medical staff members not directly involved in the treatment. The maximum time to enrol patients into the study was set to three years. The minimum number of patients was set at 30/group.

### 2.6. Statistical Analysis

Semi-quantitative data (VAS, ODI) are presented as the median (interquartile range). The VAS and ODI scores were compared among groups by means of the Kruskal–Wallis ANOVA with the post hoc Dunn test and Bonferroni correction. The correlations between the VAS or ODI score and age and body mass index (BMI) were assessed by correlation analysis (Spearman’s rank correlation coefficient). Only complete sets of data from correctly completed questionnaires were used. *p*-values less than 0.05 (*p* < 0.05) indicated statistical significance. Statistical analyses were performed with NCSS software, version 11.0.7 (NCSS, LCC, Kaysville, UT, USA).

The presented data do not have a “normal” distribution. The statistical methods used were chosen with this in mind.

## 3. Results

### 3.1. Patient Inclusion and Demographic Characteristics

The age range of the patients was 20–84 years (median age: 56 years), and there were 63 males and 115 females. The median BMI was 27.5. No statistically significant correlations were found between age, or BMI and the pre-treatment VAS or ODI score or the post-treatment changes in the scores over the follow-up period (maximum absolute values of Spearman’s rank correlation coefficient: (VAS—age: 0.0678, VAS—BMI: 0.0726, ODI—age: 0.193, ODI—BMI: 0,0843). There were no significant differences in the distribution of sides or levels of the treated nerve roots (*p* = 0.50970). Patients with failed back surgery syndrome (FBSS) accounted for 30.9% of all patients, with no significant differences among the groups (*p* = 0.13578). The demographic characteristics of the patients are summarized in Table 1.

### 3.2. Outcome Data

The VAS and ODI scores (median and interquartile range) are shown in Figure 3 and Figure 4, respectively.

#### 3.2.1. Vas Scores

The median pre-treatment VAS score of all groups was six points. At the early post-treatment time points, the largest reduction in the median VAS score (to 3.5 points) was observed in the TFESI group (a decrease of 41.7%); in the PRF and TFOOT groups, the VAS score decreased to 4 points (a decrease of 33%) and 5 points (a decrease of 16.7%), respectively. There was a significant difference in the early post-treatment VAS score between the TFESI and TFOOT groups (*p* = 0.0152). The median VAS scores at the third and sixth months after treatment were comparable among groups (five points), and these values were lower than the pre-treatment values. The persistent decrease in the median VAS score, therefore, was approximately 16.7% with respect to the pre-treatment state. There were no statistically significant differences in the VAS score among groups at three or six months after treatment.

#### 3.2.2. ODI Scores

The pre-treatment median ODI scores were 24, 20, and 23 points for PRF, TFOOT, and TFESI, respectively. At the third month after treatment, the median ODI scores in the PRF, TFOOT, and TFESI groups were 20, 18, and 20.5 points, respectively, and at the sixth month, the scores were 21, 18.5, and 18 points, respectively. The range of ODI decrease was 7.5 to 21.7%. No significant differences in the ODI score were found among the methods at any follow-up timepoint.

#### 3.2.3. Adverse Effects

Two cases of non-severe and highly transient complications (nausea and mild headache) were observed in the TFOOT group immediately after the procedure. Both patients fully recovered within 15 min, with no residual complaints. An early post-treatment increase in the VAS score was noted in 45 patients: 15 (26.3%) in the PRF group, 20 (29%) in the TFOOT group, and 10 (19%) in the TFESI group.

## 4. Discussion

This study was performed to compare the changes in pain and disability following three minimally invasive CT-guided methods, PRF, TFOOT and TFESI, in patients with chronic unilateral radicular syndrome at L5 or S1 who are resistant to conservative treatment.

Based on the published literature, we expected that the therapeutic efficacy of the newer mini-invasive methods (PRF and TFOOT) would be better than that of TFESI [14,23,24]. In our study, the overall results in the TFOOT and TFESI groups were less encouraging. We observed a reduction in pain in all three groups with respect to the baseline at all follow-up timepoints. However, PRF and TFOOT did not yield significantly better results than did TFESI. The only statistically significant difference among the treatment groups was found between the TFESI and TFOOT groups immediately after treatment, when the TFESI group showed the largest reduction in the VAS score among the groups. We did not find any significant differences in the VAS or ODI score among the three groups at the third or sixth month follow-up periods. A slight functional improvement (a decrease in the ODI score) was apparent in all groups at the third and six months follow-up period, without any significant differences.

The discrepancy of the effect of TFESI between previous studies and our study may be explained by the differences in the drug mixtures used. Another difference between these studies is the duration of pain before the mini-invasive treatment. Nonetheless, our data are in line with previous reports showing the greater efficacy of TFESI in the short term. In their systematic review, Abdi et al. summarized the effects of lumbar TFESI in managing lumbar radicular pain, showing that the evidence was strong for pain relief in the short term (six weeks or less) and moderate for pain relief in the long term (six weeks or longer) [6]. Moreover, Buenaventura et al. [25], in their systematic review, concluded that the indicated levels of evidence for lumbar TFESI in the management of lumbar nerve root and LBP was II-1 for short-term relief and II-2 for long-term improvement. In our opinion, these and our data can be explained by the prompt effect of local anaesthetic and/or corticosteroid. This hypothesis is supported by many studies showing that corticosteroids have an immediate effect on pain by inhibiting the synthesis of proinflammatory cytokines that are released locally at the injury site [26]. For these reasons, corticosteroids, in association with local anaesthetics, are actually still considered as a treatment of choice in the pain management of chronic spinal pain (cervical and lumbar) [27].

Regarding OOT, a number of studies evaluating OOT for treatment of LBP have focused on intradiscal O_2_-O_3_ therapy. This approach can be supplemented by a periforaminal O_2_-O_3_ injection [23,24,28,29,30,31]. In their systematic review, Magalhaes et al. [32] evaluated the therapeutic results of percutaneous injections of ozone for LBP secondary to disc herniation. The indicated level of evidence for long-term (more than six months) pain relief was II-3 for ozone therapy applied intradiscally and II-1 for ozone therapy applied at the paravertebral muscle and periforaminally. On the other hand, Cunha e Sa and Gonçalves [33] stated that the evidence on the safety and therapeutic advantages of intraforaminal ozone therapy over other conventional therapeutic modalities available (including completely non-invasive approaches, such as a physical therapy) for various spinal conditions is insufficient.

In our study, there were no clinically severe periprocedural complications. However, two patients in the TFOOT group complained of transient nausea and mild headache immediately after the procedure. The commonly reported minor complications of O_2_-O_3_ therapy are insomnia, itching, papules around the point of infiltration, gastritis, dizziness, tachycardia, and hot flashes [16]. Severe complications of OOT have also been reported (e.g., vertebrobasilar stroke, bilateral vitreo-retinal haemorrhages, pneumocephalus, spondylodiscitis, fulminant septicaemia) [34].

Moreover, Vanni et al. [35] described hard adhesions between the soft tissues and bony structures that were unexpectedly discovered during surgery as a possible consequence of intraforaminal OOT. Complications of PRF interventions are extremely rare, and most side effects, such as local swelling, pain at the site of needle insertion, and pain in the extremities, are transient and self-limited. More severe complications can include neural trauma, haematoma formation, and infection [21].

One limitation of this study is its inclusion of a relatively small and heterogeneous population. We recruited patients primarily on the basis of their anamnesis and clinical symptomatology when unilateral radicular syndrome was the strictly dominant complaint. A range of degenerative or postoperative spinal changes was observed on the pre-treatment MR and CT images. The use of more specific selection criteria for patients based on imaging findings may have limited the possibility of selecting a large number of patients for the study. We are also aware of the fact that the extent of blinding of this study may have been limited by the capacity of our workplace.

## 5. Conclusions

In summary, we found that TFESI (corticosteroid with local anaesthetic) was the most effective procedure among the examined treatments for reducing early post-treatment pain in patients with chronic lumbar radicular syndrome. However, contrary to our expectations, there were no significant differences in efficacy among the three evaluated methods over the long term (three and six months’ follow-up). Thus, the CT-guided mini-invasive methods PRF, TFOOT and TFESI are equally effective and can be used as complementary treatments in the management of patients suffering from chronic lumbar radicular syndrome at L5 or S1.

## Figures and Tables

**Figure 1 medicina-57-00136-f001:**
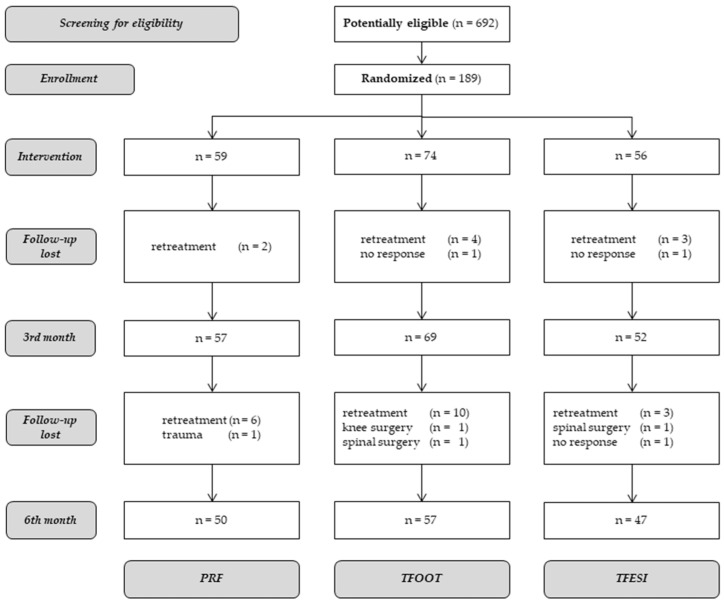
Flow diagram of the patient inclusion process.

**Figure 2 medicina-57-00136-f002:**
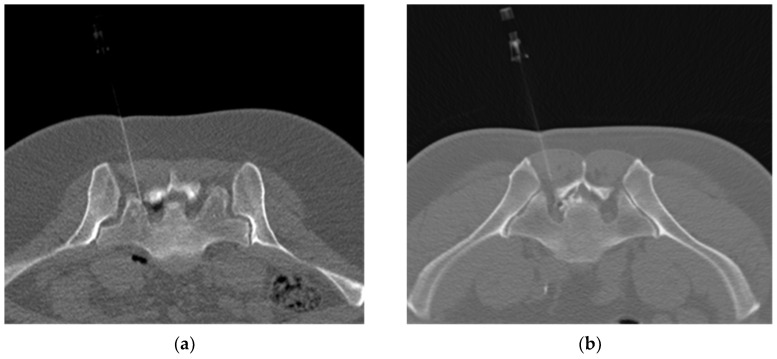
(**a**) Axial low-dose computed tomography (CT) scan showing the distribution of the O_2_-O_3_ gas mixture epidurally injected by the left-sided nerve root S1. (**b**) Axial low-dose CT scan showing the distribution of the corticosteroid and local anaesthetic mixture marked by the non-ionic contrast agent and epidurally injected by the left-sided nerve root S1.

**Figure 3 medicina-57-00136-f003:**
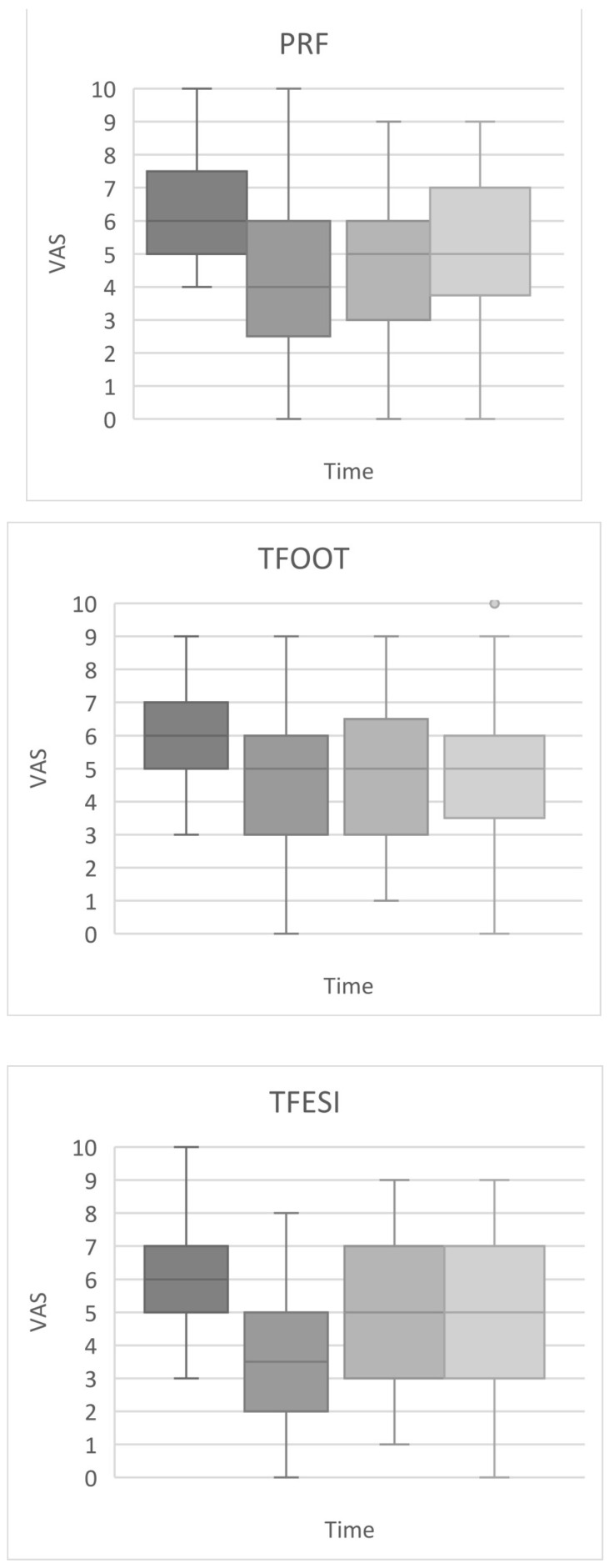
Changes in pain intensity (visual analogue scale (VAS)) over time (pre-treatment, immediately after procedure, 3rd and 6th month after procedure) figured on box plots. Footnotes: boxes indicate median, 1st, and 3rd quartile, whiskers indicate maximum and minimum levels of pain, except sporadic point outliers (in the transforaminal oxygen ozone therapy (TFOOT) group). The VAS scores were compared among groups by means of the Kruskal–Wallis ANOVA with the post hoc Dunn test and Bonferroni correction. PRF—pulsed radiofrequency; TFESI—transforaminal epidural steroid injection.

**Figure 4 medicina-57-00136-f004:**
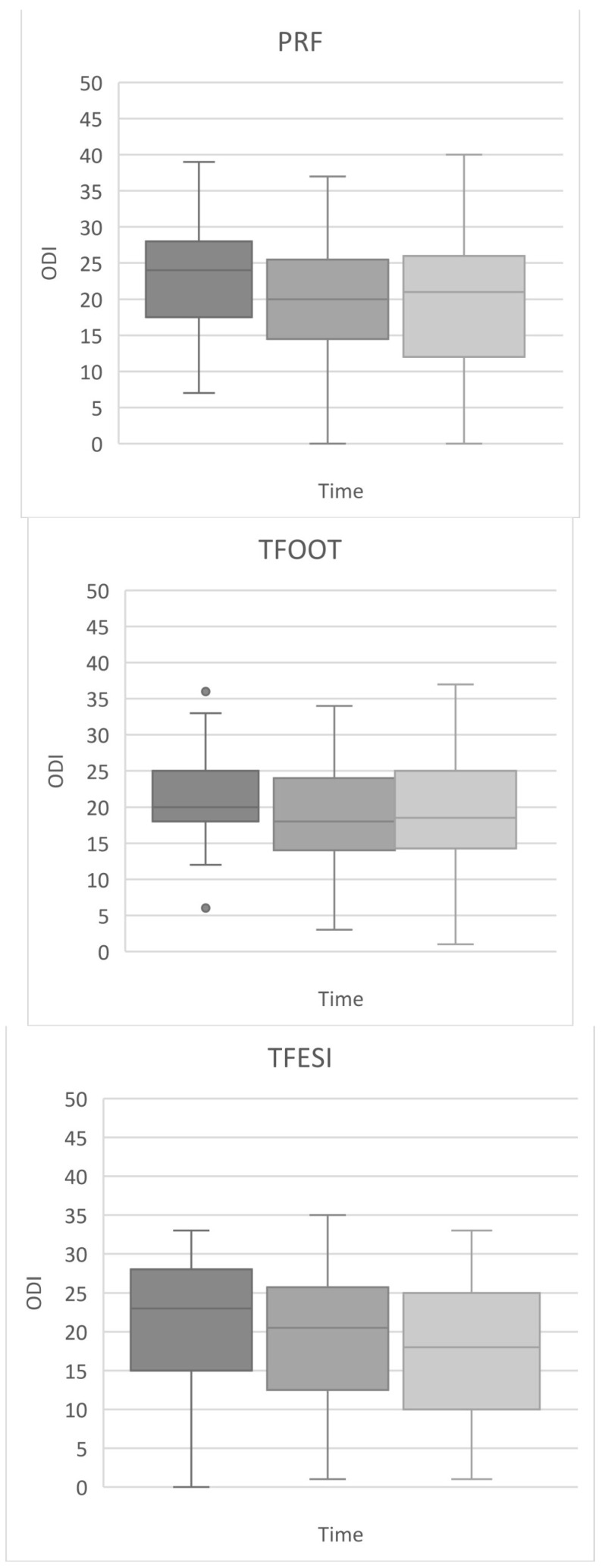
Development of disability (quantified by the Oswestry disability index (ODI)) over time (pre-treatment, 3rd, and 6th month after procedure) figured on box plots. Footnotes: boxes indicate median, 1st, and 3rd quartile, whiskers indicate maximum and minimum levels of pain, except sporadic point outliers (in the TFOOT group). The ODI scores were compared among groups by means of the Kruskal–Wallis ANOVA with the post hoc Dunn test and Bonferroni correction.

**Table 1 medicina-57-00136-t001:** Patient demographic characteristics.

Group		PRF	TFOOT	TFESI	*p*-Value
No. of patients, n (%)		57 (32%)	69 (38.8%)	52 (29.2%)	
Age (years)	range (20–84)	56.0/(63.5–48.5)	53.0/(64.5–46)	59.0/(69–43.5)	0.48580
Sex	male (n = 63)	22 (38.6%)	25 (36.2%)	16 (30.8%)	0.68276
	female (n = 115)	35 (61.4%)	44 (63.8%)	36 (69.2%)	
BMI		27.6/(31.9–25.0)	27.4/(32.4–24.5)	27.9/(31.2–24.5)	0.89583
Nerves	L5 dx./sin. S1 dx./sin.	19/14 10/14	15/15 18/21	19/13 9/11	0.50970
FBSS	yes	23 (40.4%)	20 (29.0%)	12 (23.1%)	0.13578
	no	34 (59.6%)	49 (71.0%)	40 (76.9%)	

The data are presented as a number (percent) in each group, unless otherwise specified. Age and body mass index (BMI) values are presented as the median/interquartile range (q3–q1). Data compared using Pearson’s Chi-squared test. PRF—pulsed radiofrequency; TFOOT—transforaminal oxygen ozone therapy; TFESI—transforaminal epidural steroid injection; FBSS—failed back surgery syndrome.

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
