# Peer review of "Comparison of Pulsed Radiofrequency, Oxygen-Ozone Therapy and Epidural Steroid Injections for the Treatment of Chronic Unilateral Radicular Syndrome"

_medicina, 2021, doi:10.3390/medicina57020136_

Round 1
Reviewer 1 Report
This study was performed to compare the changes in pain and disability following three minimally invasive CT-guided methods, PRF, TFOOT and TFESI, in patients with chronic unilateral radicular syndrome at L5 or S1 who are resistant to conservative treatment.
However, I see some flaws regarding the methodology or doubts that the authors need to modify. Clinical trials are experimental evaluations of new treatments to find out the usefulness of the new treatment, the mechanism of action of the new treatment, if the effectiveness is greater than other treatments already available, the side effects of the new treatment and if they are greater or less than Conventional treatment, if it outweighs the benefits to the side effects and in which patients the new treatment is more useful, for which reason, for a few years, researchers have been forced to register this type of studies to obtain their identification number. at clinical trials.gov. Was this clinical trial registered?
The authors refer to the fact that it is a single-blind study, but according to the data provided by the authors, can they explain how the patients did not know what treatment they were applying? The sensation of an electrode or needle is not the same, so it cannot be considered simple blind.
The same researcher who applied the interventions analyzed the results? This implies an important bias when analyzing the results. Not referenced in the text.
How was the randomization process carried out? With what procedure? Not explained in the text
Figure 1 does not read very well, it is not very legible.
Regarding the results, did we start from a totally homogeneous sample, the normality of the continuous variables?
If so, could they explain the statistical procedure carried out as well as the contribution of the data in results?
Author Response
Dear Reviewer,
Thank you very much for your useful and inspiring suggestions and comments, which we tried to incorporate in our manuscript. Thanks to them, our manuscript has been significantly improved. Please see the atteched file about our responses, as well as the revised manuscript.
Best regards,
Authors

Reviewer 2 Report
Dear Authors,
There is no definition of immediate effect of the treatment. (e.g. within 30 min.)
The economic comparison of these techniques would be interesting too.
Please delete XXX from the text.
Best wishes,
Reviewer
Author Response

(The authors gave the same response as above.)

Round 2
Reviewer 1 Report
Many thanks to the authors for their corrections and quick response.
Tell him that a clinical trial is an experimental evaluation of a product, substance, medicine, diagnostic or therapeutic technique that, when applied to humans, aims to assess its efficacy and safety. Studies of promising new or experimental treatments in patients are known as clinical trials. Therefore, we are facing an experimental study in patients with invasive techniques, so a record of this study should have been previously made.
According to the WHO (World Health Organization), the registration of all clinical intervention trials is considered a scientific, ethical and moral responsibility because:
There is a need to ensure that healthcare decisions are notified with the backing of all available scientific data
Difficult to make informed decisions if there is selective publication and reporting bias
The Declaration of Helsinki states that "Each clinical trial must be registered in a publicly accessible database before recruiting the first subject" http://www.wma.net/es/30publications/10policies/b3/index.html
It will facilitate the identification of similar or identical trials so that researchers and funding agents avoid unnecessary duplication
Describing ongoing clinical trials can help identify gaps in clinical research
Informing investigators and potential participants about recruitment trials can facilitate selection
Allowing researchers and clinicians to identify clinical trials in which they may have an interest could promote collaboration among researchers. The type of collaboration may include a prospective meta-analysis
Registry verification data as part of the registration process can lead to improvements in the quality of clinical trials by making it possible to identify potential problems (such as problematic randomization methods) early in the research process
Complementary to the aforementioned, the International Committee of Medical Journal Editors (ICMJE) requires registration as a condition for the publication of research results generated by a clinical study. As it is a required element for high-impact scientific publication, it is important to register the trials once they have been approved by the CEIC.